# The POLR3G Subunit of Human RNA Polymerase III Regulates Tumorigenesis and Metastasis in Triple-Negative Breast Cancer

**DOI:** 10.3390/cancers14235732

**Published:** 2022-11-22

**Authors:** Wiebke Lautré, Elodie Richard, Jean-Paul Feugeas, Hélène Dumay-Odelot, Martin Teichmann

**Affiliations:** 1Inserm U1212, CNRS UMR 5320, ARNA Laboratory, Université de Bordeaux, 33000 Bordeaux, France; 2Inserm U1218, Université de Bordeaux, 33000 Bordeaux, France; 3Inserm U1312, BRIC, Université de Bordeaux, 33000 Bordeaux, France; 4Inserm U1098, Centre Hospitalier Universitaire, Hôpital Jean Minjoz, Université de Franche Comté, 25000 Besançon, France

**Keywords:** RNA polymerase III, triple-negative breast cancer, POLR3G, androgen receptor, FOXA1

## Abstract

**Simple Summary:**

Triple-negative breast cancer (TNBC) does not express estrogen or progesterone hormone receptors and does not overexpress the HER2 transmembrane receptor. Treatment of TNBC is challenging due to the lack of these targets for specific therapies. The identification of new targets requires a better understanding of molecular mechanisms underlying the development of TNBC. Here, we show that the embryonic isoform of human RNA polymerase (Pol) III plays an important role in the development and spread of TNBC, as well as in the expression of tumor-specific proteins. Deletion of POLR3G, the specifying subunit of the embryonic isoform of Pol III in a TNBC cell line resulted in reduced tumor growth, suppressed metastasis, and induced expression of transcription factors not present in TNBC but characteristic of other types of breast cancer. Targeting POLR3G expression or processes regulated by POLR3G expression may therefore be useful for the treatment of TNBC in the future.

**Abstract:**

RNA polymerase (Pol) III transcribes short untranslated RNAs that contribute to the regulation of gene expression. Two isoforms of human Pol III have been described that differ by the presence of the POLR3G/RPC32α or POLR3GL/RPC32β subunits. POLR3G was found to be expressed in embryonic stem cells and at least a subset of transformed cells, whereas POLR3GL shows a ubiquitous expression pattern. Here, we demonstrate that POLR3G is specifically overexpressed in clinical samples of triple-negative breast cancer (TNBC) but not in other molecular subtypes of breast cancer. POLR3G KO in the MDA-MB231 TNBC cell line dramatically reduces anchorage-independent growth and invasive capabilities in vitro. In addition, the POLR3G KO impairs tumor growth and metastasis formation of orthotopic xenografts in mice. Moreover, KO of POLR3G induces expression of the pioneer transcription factor FOXA1 and androgen receptor. In contrast, the POLR3G KO neither alters proliferation nor the expression of epithelial–mesenchymal transition marker genes. These data demonstrate that POLR3G expression is required for TNBC tumor growth, invasiveness and dissemination and that its deletion affects triple-negative breast cancer-specific gene expression.

## 1. Introduction

Among cancer in women worldwide, breast cancer had the highest incidence in 2018 with 2,088,849 new cases (11.6% of all cases) and was the leading cause of death with 626,679 cases (6.6% of all deaths) [1]. Two distinct epithelial populations, luminal and basal/myoepithelial cells, are found in the mammary gland. Multiple breast cancer subtypes were identified, which are phenotypically similar to the luminal or basal populations of the normal gland. Based on protein-coding gene expression profiling, breast tumors were classified into at least three major molecular subtypes: (i) estrogen receptor positive (ER+) luminal A or B, (ii) epidermal growth factor receptor 2 positive (HER2+) and (iii) triple-negative breast cancer (TNBC) lacking expression of estrogen and progesterone nuclear receptors (ER-, PR-) and human epidermal growth factor receptor 2 amplification (HER2) [2,3]. Approximately 15% of breast cancer belongs to the TNBC group, which also covers 85% of basal breast cancer. Nearly half of TNBC patients develop distant metastases to the lung, visceral organs and brain. Median survival time after metastasis is short (13.3 months). In addition, 25% of TNBC relapse after surgery with a mortality rate of 75% within 3 months after recurrence [4]. Due to the lack of targeted therapy, today, metastatic breast cancer remains difficult to treat. Consequently, there is an urgent need for the identification of molecular mechanisms underlying the development of TNBC that can lead to the development of novel targeted therapies, particularly for the prevention and treatment of metastases.

Human RNA polymerase (Pol) III transcribes small untranslated RNAs that contribute to the regulation of gene expression (e.g., 5S rRNA; U6 snRNA; 7SK RNA; tRNAs). Regardless of their functions in regulating transcription, splicing or translation, Pol III transcripts always act as RNAs without being translated [5]. All general transcription factors (GTFs) of the Pol III transcription system have been identified, cloned and characterized [6,7,8].

Several years ago, we identified an embryonic stem cell-specific isoform of human Pol III (9; Pol IIIα) and could show that its expression is negatively regulated during stem cell differentiation and resumes upon tumoral transformation of IMR90 fibroblasts with defined genetic elements [9,10]. Expression of the human POLR3G gene, encoding the specifying RPC32α subunit of the embryonic Pol IIIα isoform, was shown to be regulated by OCT3/4 and Nanog [11]. Its suppression led to embryonic stem cell differentiation, suggesting that POLR3G is a critical driver of stemness being required for establishment and maintenance of the undifferentiated state [11,12,13]. Moreover, muscle differentiation was not only accompanied by downregulation of POLR3G expression but was also prevented by ectopic expression of RPC32α [14]. Stem cell-specific expression patterns for POLR3G were also reported in mice, but stem cell maintenance did not appear to depend on POLR3G expression [15], a difference that could be due to the lack of primate-specific Pol IIIα-dependent transcripts in mice, such as for example the SNAR-A non-coding RNA [16].

Contribution of POLR3G to tumoral transformation was suggested by its overexpression during stepwise transformation of human embryonic fibroblasts with defined genetic elements [10]. In support and extension of this finding, it was shown that POLR3G inhibition resulted in prostate cancer cell-specific proliferation arrest and cell death [17] and that POLR3G overexpression correlated with bad prognosis of transitional cell carcinoma [18].

Here, we show that POLR3G is the only component of the Pol III transcription apparatus that is significantly overexpressed in triple-negative breast cancer (TNBC) but not in other types of breast cancer. Our experimental data show that CRISPR/Cas9-mediated suppression of POLR3G in the MDA-MB231 TNBC cell line decreases colony formation in soft agar assays and invasive growth in vitro. Importantly, the POLR3G KO impairs tumor growth and metastasis of intraductal xenografts in mice. Suppression of POLR3G activates the expression of FOXA1 and androgen receptor, two key factors that are characteristic of luminal and molecular apocrine breast cancers in which they contribute to controlling hormone response and can be targeted for therapies.

## 2. Materials and Methods

### 2.1. Cell Culture

Cells were grown at 37 °C in a humidified atmosphere with 5% CO2. MCF7, MDA-MB-231, BT-549, MDA-MB-468 and MDA-MB-453 cells were maintained in DMEM (Gibco, #41966; Thermo Fisher Scientific, Illkirch-Graffenstaden, France) supplemented with 10% fetal bovine serum (Eurobio, Les Ulis, France) and 1% (penicillin (10.000U/mL)/streptomycin (10 mg/mL); Gibco #15140). BT-474 cells were grown in RPMI 1640 (Gibco #61870) supplemented by 10% fetal bovine serum, 1% (penicillin (10.000 U/mL)/streptomycin (10 mg/mL)). MCF10A cells were maintained in DMEM/F12 (Sigma-Aldrich #N4888, Saint-Quentin-Fallavier, France) supplemented with 10% horse serum, 1% (penicillin (10.000 U/mL)/streptomycin (10 mg/mL)), 0.01 mg/mL insulin (Sigma #I9278, Lezennes, France), 20 ng/mL of epidermal growth factor (Sigma #E4127, Lezennes, France), 0.5 mg/mL hydrocortisone (Sigma #H6909, Lezennes, France).

### 2.2. RPC32α Knockout by CRISPR-Cas9 Gene Editing

The genomic locus of POLR3G, encoding RPC32α, was targeted directly downstream the start codon by employing CRISPR-Cas9 in the MDA-MB231 cell line. Single-guide RNAs (sgRNAs) were designed at http://cripsr.mit.edu (accessed on 13 June 2015) and cloned into the pSpCas9(BB)-2A-Puro (PX459) plasmid (Addgene, Watertown, MA, USA). The following oligonucleotides were used: sgPOLR3G-upper strand 5′-CACCGTATAACTGGTTCTGATGGCT-3′ and sgPOLR3G-lower strand 5′-AAACAGCCATCAGAACCAGTTATAC-3′.

Briefly, MDA-MB231 cells were grown on 10 cm petri dishes to 70% confluence and transfected with sgRNA-and Cas9-expression vector using lipofectamine LTX (Invitrogen; Thermo Fisher Scientific, Illkirch-Graffenstaden, France) according to the manufacturer’s protocol. Transfected cells were enriched by the addition of 2 µg/mL puromycin for 3 days. Clonal populations were obtained from single cells in 96-well plates and examined for successful knockout of RPC32α by Western blot and Sanger sequencing. Clones were reconfirmed by sequencing after several weeks of culture.

### 2.3. RT-qPCR

For each experiment, at least three biological replicates were used for RNA extraction (Trizol; Invitrogen, Waltham, MA, USA), reverse transcription and qPCR analysis (Takyo; Eurogentec, Angers, France) as previously described [19]. Two housekeeping genes (RPL13A gene and RPL29) were used as reference. All data were standardized to the two housekeeping genes and subsequently normalized to the non-tumorigenic cell line MCF10A or to MDA-MB-231 mother cells. The primers employed are presented in Appendix A.

### 2.4. Western Blot

Nuclear extracts were derived from cell lines in midlog phase growth at maximally 70% confluence [20]. Protein concentrations were determined (Bio-Rad Protein Assay Dye Reagent Concentrate; Bio-Rad, Marnes-la-Coquette, France #5000006). Then, 30 µg of nuclear extracts was separated by SDS-PAGE and transferred onto nitrocellulose membranes (Whatman Protran BA85; Sigma-Aldrich, Saint-Quentin-Fallavier, France #WHA10402525) as described [10]. Antibodies employed: anti-POLR3G (Santa Cruz Biotechnology, Dallas, TX, USA; sc-28712), anti-POLR3GL (Atlas Antibodies, Bromma, Sweden; HPA027288), anti-beta-actin (Santa Cruz Biotechnology; sc-81178), anti-FOXA1 (Santa Cruz Biotechnology, Dallas, TX, USA; sc-101058), anti-AR (Santa Cruz Biotechnology, Dallas, TX, USA; sc-7305), anti-Flag epitope (Sigma Aldrich, Burlington, MA, USA; F1804). R.G. Roeder kindly provided antibodies directed against RPC39 and RPC62. Proteins were visualized using clarity^TM^ Western ECL (Bio-Rad, Marnes-la-Coquette, France #1705060). Beta-actin served as loading control.

### 2.5. Cell Migration and Invasion

For migration assays, cells were cultured in transwell inserts (4 10^4^ cells/insert) with polycarbonate membranes (8 μm pores; Corning Life Sciences, Borre, France #353097). DMEM supplemented with 10% fetal calf serum was used as chemoattractant. Following 4 h incubation at 37 °C, cells having migrated through membranes were stained with crystal violet and counted.

For invasion experiments, transwell membranes were coated with Matrigel matrix (corning^R^ matrigel^R^ invasion chamber, Corning Life Sciences, Borre, France). Then, 2.5 × 10^4^ cells/insert were deposited and incubated for 22 h before staining cells, having passed coated membranes. At least four random views were chosen for cell counting of each culture well under the light microscope. At least three biological replicates were analyzed.

### 2.6. Orthotopic Xenografts

The study was performed in accordance with European Community Standards of Care and Use of Laboratory Animals under level 2 containment at the University of Bordeaux (animal house authorization number A33063916). Approval for the animal experiments was granted by the Comité d’Ethique pour l’Expérimentation Animal (CEEA50) ethics committee, Bordeaux (project number no. 9816-201705051037873vs6). MDA-MB-231 and POLR3G KO MDA-MB-231 cells were transduced with a lentiviral vector conferring the expression of firefly Luciferase (MND-Luc, kindly provided by the Vectorology platform, SFR Transbiomed, Bordeaux University, Bordeaux, Nouvelle-Aquitaine, France). A total of 10,000 cells (in a volume of 10 µL), were injected through the nipple into the mammary ducts of the fourth inguinal mammary glands of female NSG mice (Jackson Laboratory, Bar Harbor, ME, USA, strain number 005557), as described previously [21]. For each experiment, 10 mice per group were injected. Tumor progression was verified once per week using a Photonimager (Biospace Lab, Nesles-la-Vallée, France). Prior to imaging, the mice were anesthetized with isoflurane. Mice were injected intraperitoneally with 100 μL of d-luciferin 150 mg/kg (Promega, France) in PBS, shaved around the tumor site and imaged during 15–30 min to reach steady-state luminescence. At the end of the experiment, some mice were sacrificed, mammary glands were removed and fixed in 4% buffered formaldehyde for IHC. Immunohistochemical staining was performed on a Benchmark-ULTRA (Ventana Medical Systems, Tucson, AZ, USA) according to the manufacturer’s recommendations on whole gland section with the following antibodies: Human-specific CK7 (SP52, Roche Ventana, 32 min, pre-diluted), Ki67 (30-9, Ventana, 32 min, pre-diluted), E-cadherin (EP700Y, Ventana, 12 min, pre-diluted), Vimentin (Cl9, Ventana (790-2917), 20 min, pre-diluted) and ER (SP1, Ventana, 32 min, pre-diluted). Immunohistochemistry was performed at the platform of experimental pathology/Institut Bergonié Bordeaux.

### 2.7. Statistical Analysis

Graphical and statistical analyses were produced on GraphPad Prism 7 (GraphPad Software INC, San Diego, CA, USA), funder H. Motulsky. Data were evaluated by ANOVA with Boferroni’post hoc analyses. *p* values are indicated in Figure legends (* *p* < 0.05; ** *p* < 0.01 and *** *p* < 0.001).

### 2.8. Transcriptome Analysis

Transcriptome analysis and figures (heatmaps, 3D scatterplots) were performed and created, respectively, using R software version 3.6.0, (R Foundation for Statistical Computing, Vienna, Austria). A total of 2627 samples (2505 primary breast tumors and 122 samples of normal breast tissue) were used after GC-RMA normalization of raw data (Affymetrix HG-U133 plus2 arrays, GPL570 platform) and suppression of samples with extreme anormal values. Raw data were downloaded from Gene Expression Omnibus (GEO accessed on 7 November 20220) and Array Express (EMBL-EBI accessed on 4 September 2020) repositories. The following series were collected: GSE12276, GSE12790, GSE16446, GSE17700, GSE18864, GSE19615, GSE20685, GSE21653, GSE22035, GSE22513, GSE23177, GSE23720, GSE25407, GSE26639, GSE30010, GSE3744 (normal breast tissue biopsies from volunteering healthy women), GSE5764, GSE6532, GSE9195, EMTAB365, ETABM854. Breast cancer samples were classified according to PAM50 gene expression signature (genefu R package) as luminal A, luminal B, HER2, Basal. For few complementary analyses, basal samples were subtyped as BL1 (basal-like 1), BL2 (basal-like 2) and M (mesenchymal) according to Lehmann’s refined classification [22].

Differential expression analysis was carried out according to the general linear model (glm R package). *p*-values were corrected for multiple comparisons by computing FDR values (false discovery rate, p.adjust R package).

## 3. Results

POLR3G has been described as a stem cell factor whose expression increases during controlled transformation of fibroblasts in vitro. Our aim here was to identify and characterize the impact of POLR3G on tumorigenesis and metastasis in a clinically relevant model.

### 3.1. POLR3G Displays Specific Overexpression in Clinical Samples of Triple-Negative Breast Cancer (TNBC)

We regrouped publicly available transcriptomic data of 2627 clinical breast cancer samples into luminal (ER+ luminal A and B tumors), basal (comprising basal-like 1, basal-like 2, mesenchymal and mesenchymal stem-like subtypes of triple-negative breast cancer), HER2-overexpressing tumors and normal breast samples according to characteristic gene expression signatures (Materials and Methods) (Figure 1) [2,22,23,24]. We analyzed mRNA expression levels of Pol III subunits and general transcription factors (GTFs) in these clinical breast cancer samples (Figure 1 and Figure 2). POLR3G showed globally low expression levels but was specifically overexpressed in triple-negative (ER-, PR-, HER2-) and more occasionally in luminal B breast cancer (Figure 1). In contrast, paralogous POLR3GL gene expression was high in all samples with maximum levels in normal breast tissue (Figure 1).

We then individually correlated the expression of different components of the transcription III system (RNA polymerase III subunits, transcription factors or regulators) to the distinct molecular subtypes of breast cancer. Individual patients were positioned in dot plot analyses relative to their expression of the Ki67 gene proliferation marker, the HER2 gene and the ESR1 (estrogen receptor) gene. The expression levels of these three genes allowed for discriminating the different molecular breast cancer subtypes from each other and from the control (basal, HER2, luminal and normal) (Figure 2). The highest POLR3GL expression was found in normal breast tissues (Figure 2A). In contrast, the highest expression of the POLR3G gene was detected in basal subtypes of breast cancer (Figure 2A). Of the other Pol III-specific subunits, POLR3E expression showed a tendency to be increased in the luminal molecular subtype (Figure 2A). Expression levels of other Pol III subunits (also those shared by Pol II or Pol I), its general transcription factors and its regulators were enhanced in either luminal or HER2-overexpressing cancers, but none tended to be specifically overexpressed in the basal breast cancer subtypes (Figure 2B–D). The expression of TFIIIB components BRF1, BRF2 and BDP1 was in tendency higher in luminal subtypes, that of most TFIIIC components (C2-C6) in Her2-overexpressing cancer, but none of these general Pol III transcription factors showed increased expression levels in basal breast cancers. Collectively, these clinical gene expression data showed that the overexpression of POLR3G in triple-negative breast cancer is not part of a general overexpression program concerning the entire Pol III transcription machinery, but rather a gene-specific regulation of this particular subunit in this type of breast cancer.

### 3.2. Characterization of RPC32α Expression in Breast Cancer Cell Lines

Breast cancer cell lines reflect many genomic and transcriptional features of primary breast tumors [27]. We analyzed POLR3G/RPC32α expression levels in two luminal (BT-474, MCF7), three triple-negative (MDA-MB-231, BT-549, MDA-MB-468), a molecular apocrine (MDA-MB-453) and an immortalized non-tumorigenic cell line (MCF10A). RT-qPCR analyses showed that POLR3G mRNA was more than three-fold overexpressed in all triple-negative breast cancer cell lines (TNBC-CL) whereas POLR3GL expression levels were below those of the non-tumorigenic control (Figure 3A). Western blot analyses confirmed strong RPC32α expression in the three TNBC-CL, lower levels in the MDA-MB-453 cell line and very low expression in all other cell lines including the control (Figure 3B). In contrast, the highest RPC32β expression was observed in the non-tumorigenic MCF10A cell line and lower levels found in the other breast cancer cell lines. POLR3C mRNA expression levels were globally equivalent to the non-tumorigenic MCF10A control, except for the BT-549 TBNC cell line (Figure 3A). RPC62 protein levels were slightly higher in all analyzed breast cancer cell lines than in the MCF10A cell line (Figure 3C). These results were in agreement with the clinical transcriptomic data (Figure 1 and Figure 2A), showing specifically increased POLR3G mRNA expression levels in basal breast cancer (including TBNC).

### 3.3. CRISPR/Cas9-Mediated KO of POLR3G/RPC32α in the MDA-MB-231 TNBC Cell Line

To analyze whether RPC32α overexpression in triple-negative cell lines simply accompanied tumorigenesis or whether it had decisive functions in regulating growth, transformation and generation of metastases, we generated POLR3G knockout (3GKO) cell lines in MDA-MB-231 cells by employing CRISPR-Cas9 (Materials and Methods). We selected three homozygous KO clones, one of which contained a two-nucleotide deletion of the thymine and the guanine contributing to the start codon (KO RPC32α-1) and two others (KO RPC32α-2 and -3) missing a guanine downstream of the start codon (Appendix A). Western blot analyses confirmed that the three clones expressed neither RPC32α (Figure 4A, panel to the left) nor Cas9 (Appendix A). We chose clones that did not stably integrate Cas9 in order to avoid Cas9-mediated off-target effects. Furthermore, 3GKO cell lines showed POLR3G and POLR3GL mRNA levels slightly above those in the MDA-MB231 mother cell line (Figure 4A, panel to the right). Moreover, comparable RPC32β protein expression levels in MDA-MB231 and 3GKO cells were determined by Western blot (Figure 4A, panel to the left), indicating that POLR3GL/RPC32β expression was not increased to compensate for the loss of POLR3G/RPC32α.

### 3.4. The POLR3G/RPC32α KO Neither Alters the Expression of Polymerase Subunits Nor Pol III Transcription

As RPC32α represents one of the seventeen Pol III subunits, we analyzed next whether its suppression affected the expression of other Pol III subunits (POR3C, POLR3D and POLR3F). RT-qPCR revealed that neither POLR3C, POLR3D and POLR3F mRNA expression levels (Figure 4A, panel to the right) nor RPC62 (encoded by POLR3C) and RPC39 (encoded by POLR3F) protein levels (Figure 4B) significantly differed in the three 3GKO cell lines from those of the parental MDA-MB-231 cell line.

Likewise, RT-qPCR analyses showed that POLR3G KO did not impact on Pol III transcription of representative genes directed by distinct Pol III promoter types (for review of Pol III promoter types, see [8,28]) (Figure 4C), suggesting that these RNAs were readily synthetized by the POLR3GL/RPC32β-containing Pol IIIβ isoform in the absence of RPC32α-containing Pol IIIα.

### 3.5. RPC32α Expression Regulates Anchorage-Independent Growth of the MDA-MB231 Breast Cancer Cell Line In Vitro

Microscopic cell counting and MTT assays revealed no differences in 2D growth and viability between the parent MDA-MB-231 and 3GKO cell lines (Appendix A). In contrast, anchorage-independent 3D growth in soft agar assays as an indicator of tumorigenicity in vitro demonstrated that 3GKO cell lines formed on average 89% less colonies than MDA-MB-231 cells (9.7 ± 3.1%, 5.7 ± 1.6% and 18.6 ± 2.6% mean colonies for KO RPC32α-1, KO RPC32α-2 and KO RPC32α-3, respectively, compared to MDA-MB-231wt cells (Figure 5A, *p* < 0.01)). In addition, the size of the few residual colonies was also diminished in 3GKO cells lines. The mean diameter of RPC32α knockout cell colonies ranged from 54 to 79 µm in contrast to an average of 91 µm in MDA-MB231 wild-type cells. Thus, the calculated mean volume of the remaining 3GKO colonies was 21–65% of that formed by the MDA-MB231 cell line ((mean diameter of 3GKO colonies)^3^ divided by (mean diameter of MDA-MB231 colonies)^3^). Since 2D cell growth is not reduced in 3GKO clones, the impaired colony formation is likely caused by other mechanisms than growth control only. Collectively, these data suggested a growth regulation-independent contribution of RPC32α expression to colony formation of triple-negative breast cancer cells in vitro.

### 3.6. The RPC32α Knockout Decreases Invasive but Not Migratory Capacity of Triple-Negative Breast Cancer Cells In Vitro

Next, we analyzed cell migration and invasive capacities in Boyden chamber assays. Appendix A shows that all the cell lines (expressing RPC32α or not) were able to move across porous membranes toward the 10% serum containing medium at similar rates. Thus, the loss of RPC32α expression did not change migratory capacities in vitro. To mimic conditions of infiltration and dissemination, we determined the capacity of cells to migrate through matrigel-coated Boyden chambers. The number of 3GKO cells invading the lower medium-containing chamber was reduced by around 89% compared to the MDA-MB-231 cell line (6.2 ± 0.6%, 17.2 ± 1.3% and 9.9 ± 1.1% mean number of invaded cells for KO RPC32α-1, KO RPC32α-2 and KO RPC32α-3 cell lines, respectively, compared to the MDA-MB-231wt cell line (Figure 5B *p* < 0.0001)). Thus, POLR3G/RPC32α KO efficiently reduced invasion through matrigel but did not interfere with migration in vitro, suggesting that POLR3G may fulfil important roles in the process of TNBC dissemination and metastasis formation in vivo.

### 3.7. The RPC32α Knockout Impairs Triple-Negative Breast Tumor Growth In Vivo

We then aimed to assess in vivo contributions of RPC32α to tumor development upon xenotransplantation into mice. For that purpose, we established bioluminescent MDA-MB-231 wild-type and 3GKO cell lines in RPC32α KO clones 1 and 2. Luminescence was proportional to the number of cells and was detectable for as little as 40 cells in all cell lines, indicating that tumor development could be followed from the earliest stages onward. We confirmed that the addition of the luciferase gene in the different cell lines did neither change RPC32α expression nor modify soft agar growth properties in vitro (Appendix A).

Orthotopic injections of cells into the milk duct of immunodeficient mice (NSG) (Materials and Methods) showed that 3GKO cell lines developed tumors with an average delay of two weeks compared to the MDA-MB-231 parental cell line. As a consequence, four weeks after injection, the RPC32α KO caused a significant decrease in tumor size in both knockout cell lines (Figure 6A, B; *p* < 0.001), being consistent with the observations of reduced colony formation in soft agar assays in vitro (Figure 5A). Hematoxylin and eosin (HE) staining of resected tissues showed that MDA-MB-231 cells invaded almost all of the mammary gland whereas 3GKO cells produced small tumors (Figure 6C, upper panel). Immunohistological staining with the human-specific cytokeratin-7 (CK7) antibody (Figure 6C, middle panel) perfectly overlapped with the HE staining of tumor cells, strongly suggesting that these tumors were of human origin. These results demonstrated that the RPC32α knockout cells were less capable of forming tumors in nude mice than parental MDA-MB-231 cells.

### 3.8. POLR3G/RPC32α KO Strongly Reduces Metastatic Dissemination of Triple-Negative Breast Cancer In Vivo

Since the KO of RPC32α led to reduced invasive capacity of 3GKO cells in vitro (Figure 5B), we analyzed its impact on metastatic growth in nude mice. We repeated orthotopic xenografts with the MDA-MB-231 and the two 3GKO cell lines and removed primary tumors when they reached comparable sizes as determined by luminescence of around 2 × 10^7^ ph/s/cm^2^/sr. Three weeks after resection, we analyzed the generation of metastases in the three groups. The overall luminescence in the two knockout groups was significantly lower than that of the wild-type group, indicative of lower metastasis formation in vivo (Figure 7A). In particular, lungs, kidneys, ganglia and the pancreas contained less metastases in 3GKO xenografts than in MDA-MB-231 wt xenografts (Figure 7A, right panel and Figure 7B, the location of the organs is indicated by arrows in the image shown on the right). These results demonstrate that the POLR3G/RPC32α KO in the MDA-MB-231 cell line strongly reduced the formation of metastases.

### 3.9. The POLR3G/RPC32α KO Neither Changes EMT Nor Proliferation Marker Expression

Because the 3GKO clones exhibited attenuated tumor development in mice, we analyzed the expression of molecular markers for different breast cancer types by immunohistochemical staining in resected tumors. Appendix A shows representative staining of tissues isolated from mice xenotransplanted with MDA-MB231 (three squares to the left in all rows) or 3GKO cells (three squares to the right in all rows). All tumors (expressing or not RPC32α) were estrogen receptor negative (ER-; Figure 6C, lowest panel) and remained proliferation marker Ki67 high (Appendix A, uppermost row), indicating that the 3GKO clones did not change ER expression levels and retained proliferative capacities comparable to the parental triple-negative breast cancer cell line. Moreover, high levels of the cancer stem cell marker CD44 [29] and of the mesenchymal marker vimentin [30], as well as low levels of the epithelial marker E-cadherin were detected in resected tumor tissues generated from xenografted MDA-MB231 and from 3GKO cells (Appendix A, three rows to the bottom). Taken together, POLR3G/RPC32α knockout cell lines showed strongly reduced tumor and metastasis development compared to the MDA-MB-231 cell line, but POLR3G suppression did not change the expression of several principal basal breast cancer marker proteins.

### 3.10. Expression of RPC32α Is Negatively Correlated with FOXA1 and AR Expression in TNBC Patients

When analyzing the transcriptomic data of clinical cases demonstrating POLR3G overexpression in basal breast cancer (Figure 1), we determined the genes being positively or negatively correlated to POLR3G expression in breast cancer patients. None of the protein-coding genes showed a very strong positive or negative correlation (correlation value > 0.7) with POLR3G/RPC32α expression (Appendix A). However, negative correlations slightly below this threshold were observed for several genes involved in steroid hormone receptor-dependent gene expression. The top eight genes that negatively correlated to POLR3G expression (MLPH, FOXA1, SPDEF, AR, SIDT1, AGRP2, XBP1, GATA3) are typically overexpressed in ER+ breast tumors as compared to ER- breast tumors [27,31,32]. Therefore, we analyzed whether FOXA1 expression was influenced by the POLR3G KO.

### 3.11. The POLR3G/RPC32α Knockout in the MDA-MB-231 Cells Induces FOXA1 and AR Expression

As shown in Figure 8A,B, left panels, FOXA1 mRNA and protein expression were strongly increased in POLR3G/RPC32α KO (3GKO) cell lines. Likewise, cell extracts derived from different resected primary tumors confirmed the reactivation of FOXA1 expression in 3GKO cell lines (Figure 8D, left panel). We determined that POLR3G/RPC32α expression was unchanged in the resected tumors (Figure 8C).

FOXA1 expression is characteristic of luminal and apocrine breast tumors [32,33] and was shown to be co-regulated with androgen receptor (AR) expression [32]. FOXA1 is a pioneer transcription factor [34] that regulates luminal differentiation of breast epithelial cells [35]. As a pioneer factor, dependent on its binding to closed chromatin, FOXA1 prepares genomic landscapes for the binding of liganded estrogen and androgen receptors [36]. In ER-/AR+ molecular apocrine breast cancer cells, FOXA1 binding sites overlap with 98% of AR binding sites [37]. Since the MDA-MB-231 cell line only expressed low levels of AR, we analyzed a possible regulation of AR mRNA and protein expression in the RPC32α knockout cell lines. Comparable to FOXA1 expression (Figure 8A,B,D, left panels), we also observed increased AR mRNA and protein expression levels in both nuclear extracts derived from 3GKO cell lines and in tumors formed from 3GKO cells (Figure 8A,B,D, panels to the right). Collectively, these results suggested that the KO of RPC32α activated both expression of the pioneer factor FOXA1 and that of the androgen receptor. The fact that we found these results in clinical data as well as in vitro in the KO cell line model and in vivo in the mouse model suggests that POLR3G/RPC32α expression contributes to the regulation of FOXA1 and AR expression, thereby potentially affecting luminal or basal cell identities in breast cancer.

## 4. Discussion

RNA polymerase III transcription was shown to substantially contribute to tumorigenesis. Direct interactions of Pol III transcription factors with tumor-regulating proteins (TP53, RB, c-MYC) were suggested as possible underlying growth regulatory mechanisms [7,38,39,40]. However, Pol III transcription was not shown to regulate the expression of oncogenic drivers or differentiation factors. Consequently, it is believed that Pol III provides small RNAs supporting cell growth [41,42] but that it does not interfere with decisions that influence cell fate or differentiation.

Most published studies addressed oncogenic effects of POLR3G/RPC32α in cultured cells and described correlations between its overexpression and cancer development [10,43]. However, POLR3G was also identified in clinical data as one out of five genes in a risk signature predicting the prognosis of patients with hepatocellular carcinoma [44], significantly upregulated in transitional cell carcinoma [18], and increased POLR3G expression was detected in prostate tumor biopsies [17].

Based on gene expression profiling, our data demonstrate that POLR3G is specifically overexpressed in ER-, PR-, HER2-, triple- as well as additionally AR- quadruple negative breast cancer (Figure 1 and Figure 2), whereas the expression of other Pol III subunits or GTFs is not altered in TNBC (Figure 2). Specific overexpression of POLR3G but not of the paralogous POLR3GL subunit or any other component of the Pol III transcription system in TNBC is indicative of unique tumor-promoting functions of POLR3G in TNBC. This hypothesis was also supported by the fact that functions fulfilled by POLR3G in regulating TNBC tumor development and metastasis could not be compensated by POLR3GL expression. The fact that the expression of Pol III-transcribed genes (Figure 4C) and 2D growth on petri dishes (Appendix A) were unaffected by POLR3G KO clearly indicated that the loss of RPC32α expression did not primarily act by restricting cell growth, but rather by intervening with other regulatory mechanisms driving tumorigenesis. Data obtained in a cellular model system of stepwise tumoral transformation of human fibroblasts were also in support of POLR3G functions in tumorigenesis beyond regulation of cell growth, as in this cellular system, tumorigenesis did not coincide with strongly increased Pol III transcription levels [10], indicating that growth-supportive function of Pol III may not be the only contribution of this transcription system to tumoral transformation.

Since delayed formation of tumors and the strong reduction of metastasis formation by 3GKO cell lines compared to MDA-MB-231 cells could not be attributed to altered cell growth only, it seemed reasonable that the suppression of POLR3G qualitatively changed fundamental biological parameters of MDA-MB-231 cells. Our data indicate that altered interactions with the environment in 3GKO cell lines may underlie reduced colony formation in soft agar assays, impaired invasion, delayed tumor formation in mice and the unexpected inhibition of metastasis generation. Deletion of POLR3G activated the expression of luminal fate-specific proteins FOXA1 and androgen receptor (AR). FOXA1 is not only required for developmental specification, but also for postnatal differentiation of breast epithelial cells into luminal cell types [45,46]. We also noticed that the expression of GATA3 was negatively correlated to the expression of POLR3G in clinical data (Appendix A). Thus, it is possible that POLR3G expression influences the expression of pioneer factors, which in turn affect cell fate decisions. Our data support that, in the case of TNBC, suppression of POLR3G redirects, at least in parts, the gene expression signature of TNBC toward the one that was described for molecular apocrine breast cancer [47,48].

Our data show that the expression of POLR3G in breast cancer samples was negatively correlated with that of eight out of ten genes (GATA3; XBP1; AGR2; SIDT1; AR; SPDEF; FOXA1; MLPH; Appendix A) in a list of signature genes most differentially expressed in luminal A compared to basal-like tumors [27]. All ten genes that negatively correlated to POLR3G expression were also found in oncomine breast cancer microarrays as being co-regulated with FOXA1. Moreover, nine of the ten genes (except PRR15) were likewise co-regulated with ER expression [32]. However, only the four transcription factors in this list (FOXA1; AR; XBP1 and GATA3) were shown to be upregulated in ER+ versus ER- tumors and in the ER+ MCF7 and T-47D luminal breast cancer cell lines relative to the ER- MDA-MB231, MDA-MB435 and SK-BR-3 cell lines [49].Nevertheless, none of the genes that correlated best with POLR3G expression were found in gene expression signatures of basal-like tumors. Interestingly though, POLR3G itself was included into a signature of classifier genes expressed in basal B breast cancer cell lines [27,50]. Integrating our data on POLR3G KO-induced expression of FOXA1 and AR and the list of genes that are coregulated with POLR3G expression into published data describing genes that are coregulated with FOXA1 and ER suggests that POLR3G may be involved in regulatory functions that set gene expression patterns for hormone response.

AR expression in breast cancer is a disease prognostic marker. While AR expression is associated with favorable disease outcome in ER-positive luminal breast cancer [46,51], conflicting results concerning disease prognosis were published with respect to AR expression in TNBC [52,53]. It was suggested that up to 50% of TNBC might rely on AR activity even if expression levels are low. It was proposed that AR expression increased in cells that were grown in suspension and that AR expression possibly favored a cancer stem cell (CSC)-like population, which could be therapeutically targeted by antiandrogens such as enzalutamide [54]. The results in our cellular context, however, suggest that POLR3G suppression inhibited anchorage-independent growth while increasing AR expression levels. Moreover, POLR3G has high expression levels in embryonic stem cells, and only faint levels can be detected after stem cell differentiation [9,11], supporting the hypothesis that suppression of POLR3G may induce a more differentiated phenotype rather than producing a stem cell-like population. Given the conflicting results on AR expression and its impact on tumor outcome [46,51,53], it seems plausible that subtle differences in gene expression may explain the observed differences.

The here presented data identify POLR3G/RPC32α as an important regulator of TNBC tumor growth and generation of metastases in vitro and in mouse xenografts. Therefore, monitoring the expression of POLR3G could be useful for the diagnosis and classification of TNBC as well as for the assessment of therapeutic success. Moreover, RPC32α may represent a potential therapeutic target rendering TNBC AR-positive and thus treatable by androgen synthesis inhibitors and/or antiandrogens. Since the paralogous POLR3GL subunit is able to sustain cell growth in the absence of RPC32α, therapies targeting POLR3G may be very well tolerated.

## 5. Conclusions

Triple-negative breast cancer (TNBC) represents 15% of all breast cancers. Due to the lack of molecular targets, treatment of TNBC remains challenging. Here, we show that POLR3G/RPC32α expression regulates tumor formation and dissemination in vitro and in vivo. KO of POLR3G in the MDA-MB231 TNBC cell line reduces colony formation and cell invasion in vitro. Furthermore, it impairs tumor formation and suppresses dissemination of metastases in mouse xenografts. POLR3G expression is negatively correlated to FOXA1 and androgen receptor expression, and its deletion reactivates FOXA1 and AR expression. Thus, we conclude that POLR3G expression regulates tumorigenesis and the expression of genes that are involved in cell fate decisions in triple-negative breast cancer. Further work will show whether POLR3G may emerge as a therapeutic target in TNBC.

## Figures and Tables

**Figure 1 cancers-14-05732-f001:**
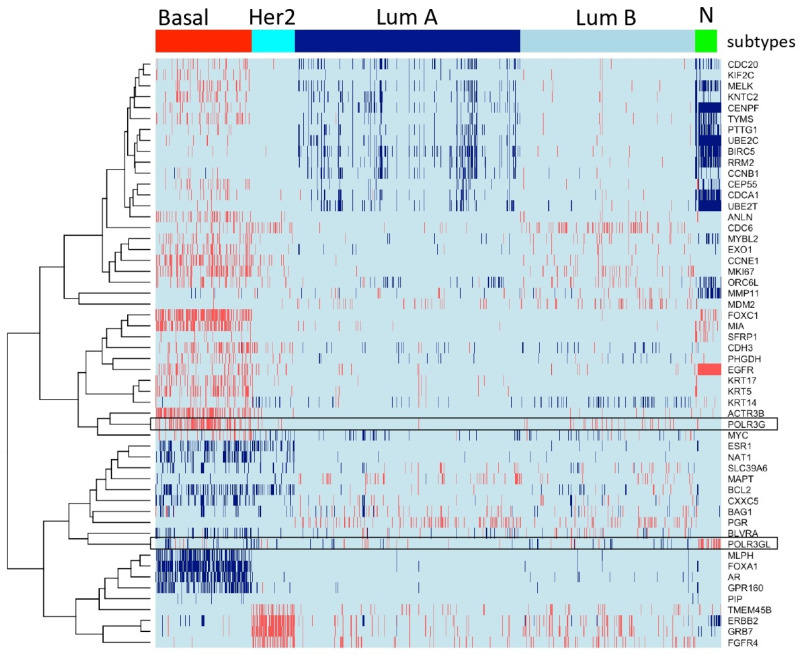
POLR3G is specifically overexpressed in samples derived from basal breast cancer patients. Heatmap representing 2627 clinical samples of breast cancer and controls (columns). PAM50 genes according to Lehmann’s refined classification [22] are appropriately in rows and are used for clustering samples according to their gene expression profiles. The expression profiles of POLR3G and POLR3GL across samples are shown in boxed rows. High transcript levels are marked in red, and low levels are marked in blue. Transcriptomes were collected from public repositories (see Materials and Methods). PAM50 classification identified the following distribution of subtypes: 445 basal (including triple-negative), 202 Her2, 1046 luminal A (LumA), 812 luminal B (LumB) tumors and 122 normal breast tissue samples (N). Colors correspond to levels of mRNA. POLR3G mRNA expression is higher in the basal group than in other subtypes. POL3GL mRNA expression is higher in normal breast tissues than in breast cancer samples (*p* value < 10^−100^; see Appendix A).

**Figure 2 cancers-14-05732-f002:**
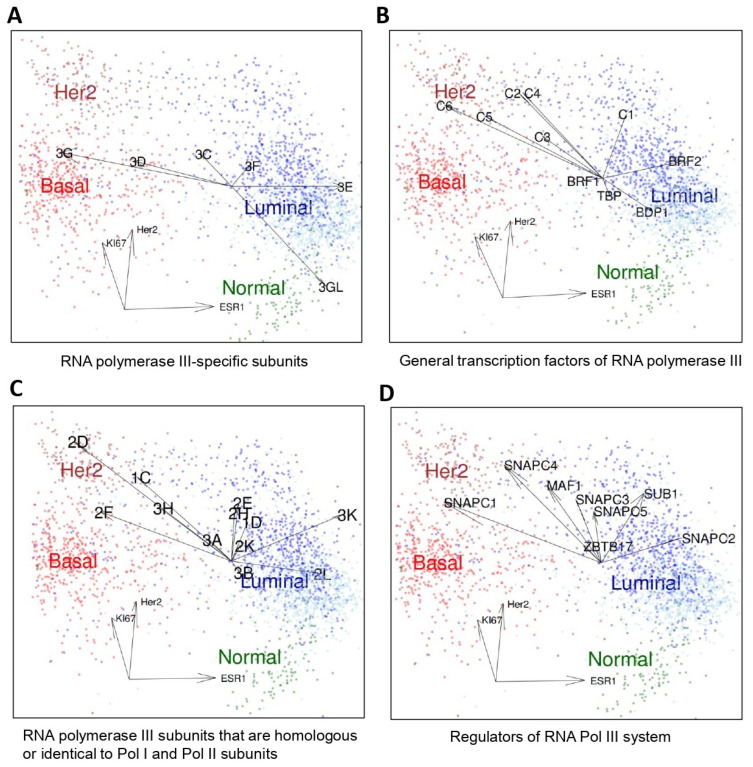
Three-dimensional scatterplot representations of the expression of Pol III subunits, transcription factors and regulators. The same 2627 samples shown in Figure 1 are presented according to the expression of HER2 (ERBB2), KI67 and ESR1 (estrogen receptor). A color code permits identification of individual tumor types and normal control: basal (red), Her2 (Brown), luminal A (dark blue) and luminal B breast cancers (light blue) as well as normal tissue samples (green). Correlations of the expression of Pol III transcription system components (polymerase subunits, general transcription factors, regulators) with these three axes are indicated by arrows. (**A**) RNA polymerase III-specific subunits. POLR3G expression correlates with the basal group and POLR3GL expression with normal tissue. The following abbreviations are used: 3G, POLR3G; 3D, POLR3D, 3C, POLR3C, 3F, POLR3F, 3E, POLR3E, 3GL, POLR3GL. (**B**) General RNA polymerase III transcription factors TFIIIC and TFIIIB. C1 to C6 correspond to GTFC1 to GTFC6 [reviewed in 6–8]. BRF1, BRF2, BDP1 and TBP constitute the two TFIIIB-α and TFIIB-βactivities. (**C**) RNA polymerase (Pol) III subunits that are homologous or identical to Pol II or Pol I subunits (as in (**A**), “POLR” has been omitted before the gene symbols for the sake of simplicity). (**D**) Regulators of RNA polymerase III system. SNAPC1-5: subunits of PTF/SNAPC complex (6–8). SUB1 = PC4 [25]. ZBTB17 = MIZ1 [26].

**Figure 3 cancers-14-05732-f003:**
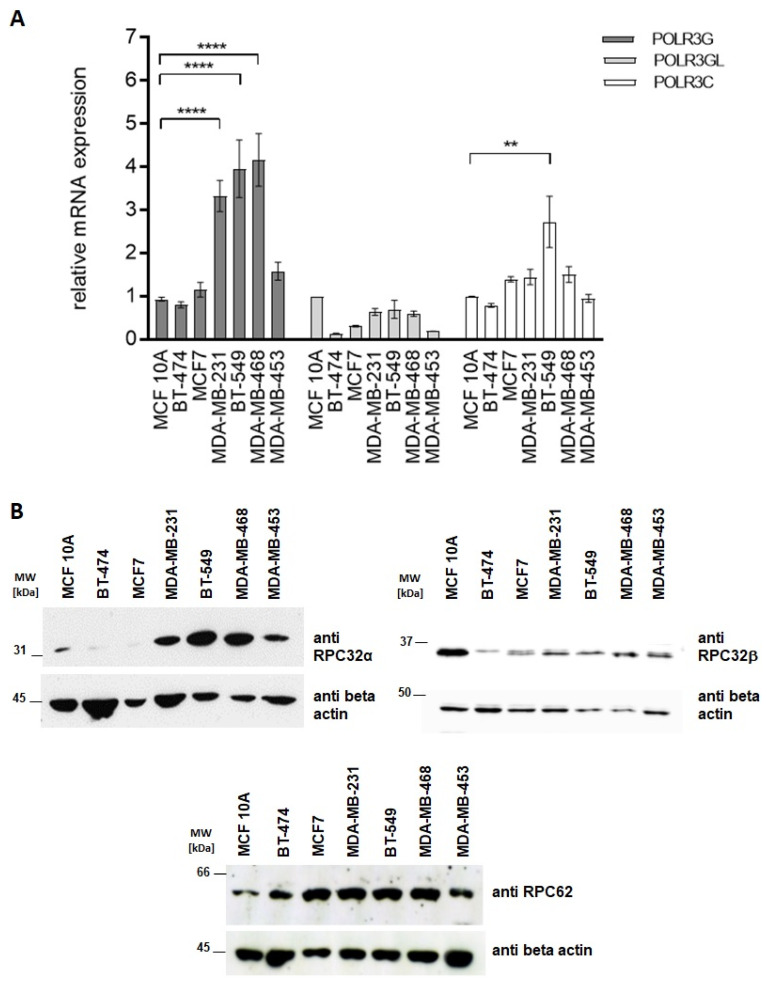
Analyses of Pol III subunit expression in breast cancer cell lines. (**A**) POLR3G mRNA is overexpressed in triple-negative breast cancer cell lines. Determination of Pol III subunit mRNA levels by RT-qPCR (Materials and Methods). Total RNAs were isolated from different breast cancer cell lines and compared with the non-tumorigenic control MCF10A cell line. Analyzed cell lines are appropriately depicted below the graphs. The code for analyzed genes is shown to the upper right. Relative mean expression of three independent experiments ± SEM (** *p* < 0.01, and **** *p* < 0.0001) is shown. (**B**) RPC32α protein is overexpressed in triple-negative breast cancer cell lines and RPC32β in the non-tumorigenic cell line. Western blot (Materials and Methods) with anti-RPC32α, -RPC32β and -beta-actin antibodies. Analyzed cell lines are appropriately depicted above the ECL images. The uncropped Western Blot images of Figure 3 can be found in Appendix A.

**Figure 4 cancers-14-05732-f004:**
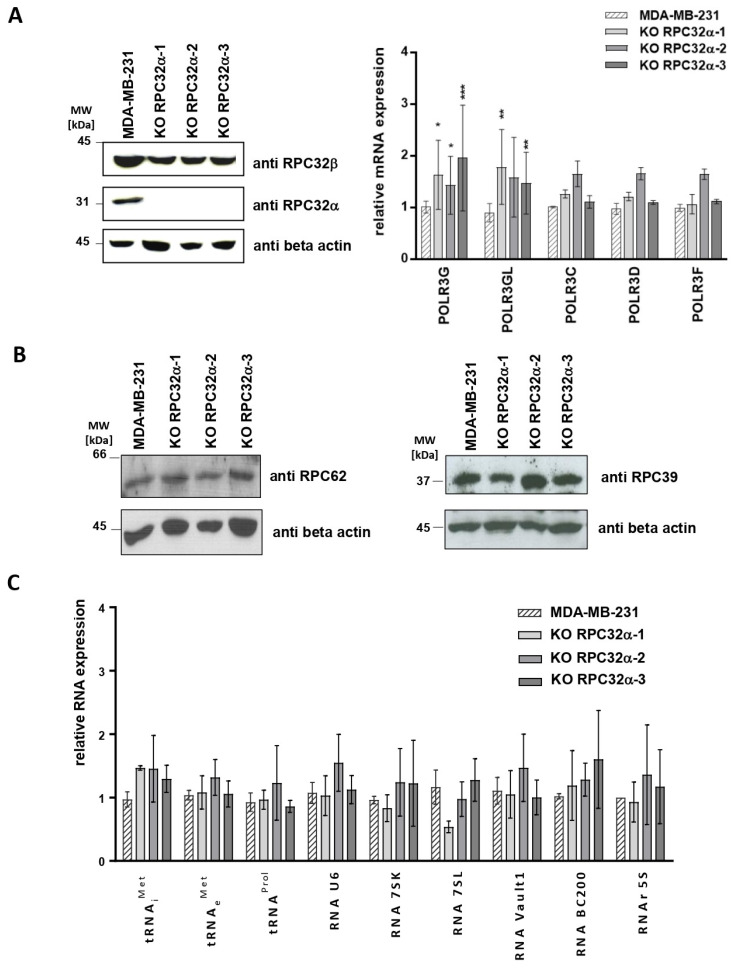
Molecular characterization of POLR3G KO (3GKO) clones. (**A**) Analysis of POLR3G KO (3GKO) clones by Western blot (to the left) with anti-RPC32α, -RPC32β and -beta-actin antibodies (Materials and Methods). Detected proteins are depicted to the right, the MDA-MB231 and 3GKO cell lines that were analyzed are appropriately indicated above the ECL images. Determination of Pol III subunit mRNA levels (to the right) in MDA-MB231 and 3GKO cell lines derived therefrom by RT-qPCR (Materials and Methods). Analyzed genes are indicated below the graphs. Relative mean expression levels of three independent experiments ± SEM (* *p* < 0.05, ** *p* < 0.01, and *** *p* < 0.001) are shown. (**B**) Western blot analyses of RPC62 (left panel) end RPC39 expression (right panel) in MDA-MB231 and 3GKO cells. Analyzed cell lines are appropriately depicted above the ECL image. (**C**) RT-qPCR analyses of the expression of RNA polymerase III-transcribed RNAs in MDA-MB231 cells and the three 3GKO clones. Relative mean expression levels of three independent experiments ± SEM (* *p* < 0.1, ** *p* < 0.01 and *** *p* < 0.001 compared with MDA-MB-231) are shown. The uncropped images of Western Blots shown in Figure 4A,B can be found in Appendix A.

**Figure 5 cancers-14-05732-f005:**
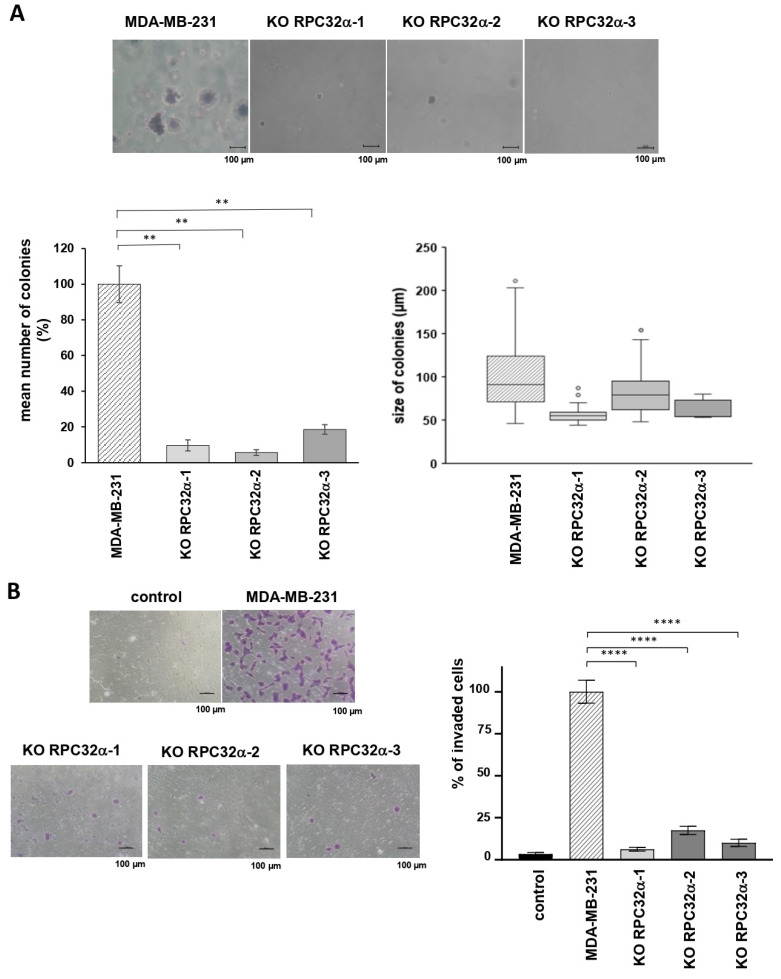
The POLR3G/RPC32α knockout decreases tumorigenic and invasive capacities of MDA-MB-231 breast cancer cells in vitro. (**A**) Soft agar assays of MDA-MB231 and POLR3G KO (3GKO) cell lines derived therefrom. The top row shows representative microscope images of colonies stained with crystal violet. The graphs in the bottom row show relative mean numbers (left) and the mean size (right) of colonies formed by MDA-MB-231 and 3GKO cell lines (three independent experiments). (**B**) Transwell cell invasion assays with MDA-MB231 and 3GKO cell lines derived therefrom. Representative microscope images are shown to the left. Scale bar: 100 µm. Percentage display of colony quantifications is shown to the right. Data are presented as the mean of three independent experiments ± SEM (** *p* < 0.01 and **** *p* < 0.0001 compared with MDA-MB-231).

**Figure 6 cancers-14-05732-f006:**
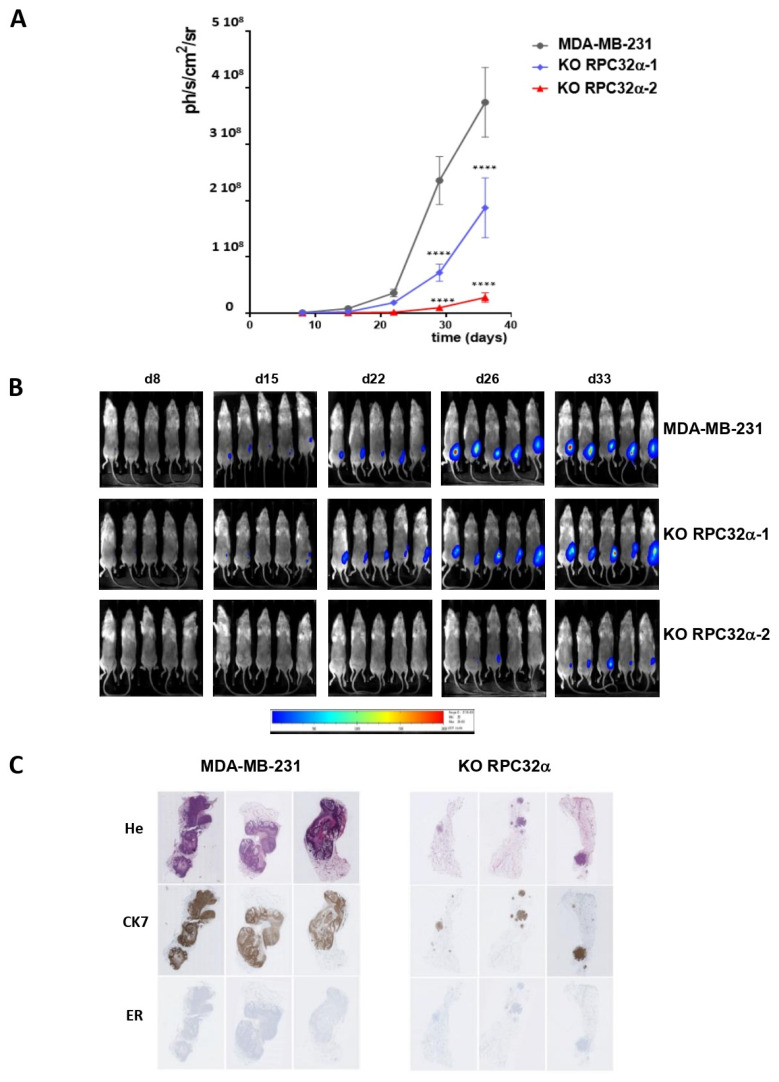
Orthotopic xenografts of MDA-MB-231-derived RPC32α KO cells show reduced primary tumor growth and metastasis formation. (**A**) Time course for mammary tumor growth of orthotopically xenografted MDA-MB231 and POLR3G KO (3GKO) cell lines as appropriately indicated. Tumor bioluminescence was measured using a Photoimager and normalized to the starting value one week after injection. Logarithmic representation of mean luminescence of two independent experiments with ten mice each (**** *p*  <  0.0001 compared to MDA-MB-231). (**B**) Representative bioluminescence imaging on live animals at days 8, 15, 22, 26 and 33 (as indicated above the images) post-orthotopic xenograft for each group. Mice that were xenografted with the MDA-MB231 cell line are shown in the uppermost row, and those that were xenografted with the two distinct POLR3G KO (3GKO) cell lines are shown in the lower two rows, as appropriately indicated to the right. (**C**) Overall morphology of formalin and paraffin embedded tumors analyzed by H&E staining (uppermost row). Immuno-histochemical staining of the same samples with anti-human cytokeratin 7 (CK7; middle row) or anti-estrogen receptor antibodies (ER; row to the bottom) as indicated to the left.

**Figure 7 cancers-14-05732-f007:**
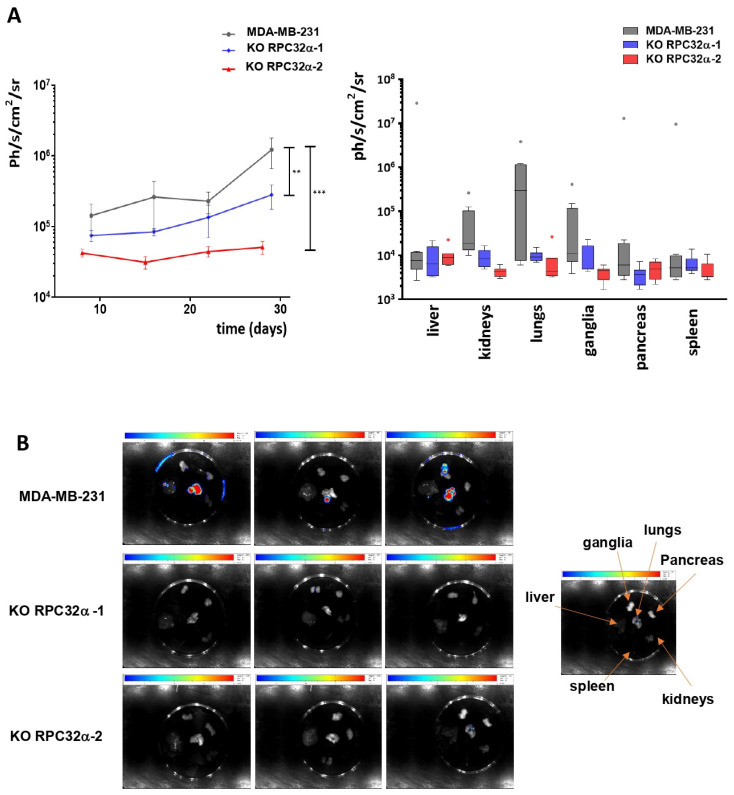
Analyses of metastases formed by MDA-MB-231 and POLR3G KO (3GKO) cell lines. (**A**) To the left, logarithmic representation of overall metastases bioluminescence ± SEM (** *p* < 0.001; *** *p* < 0.001) (MDA-MB-231: number of mice analyzed (*n*) = 9, KO RPC32α-1: *n* = 7, KO RPC32α-2: *n* = 7). To the right, logarithmic representation of metastases luminescence in distinct organs (liver, kidneys, lungs, ganglia, pancreas and spleen, as shown below the graphs). (**B**) Representative images of bioluminescence imaging of different organs 4 weeks after resection of primary tumor. The localization of organs on images is shown to the right.

**Figure 8 cancers-14-05732-f008:**
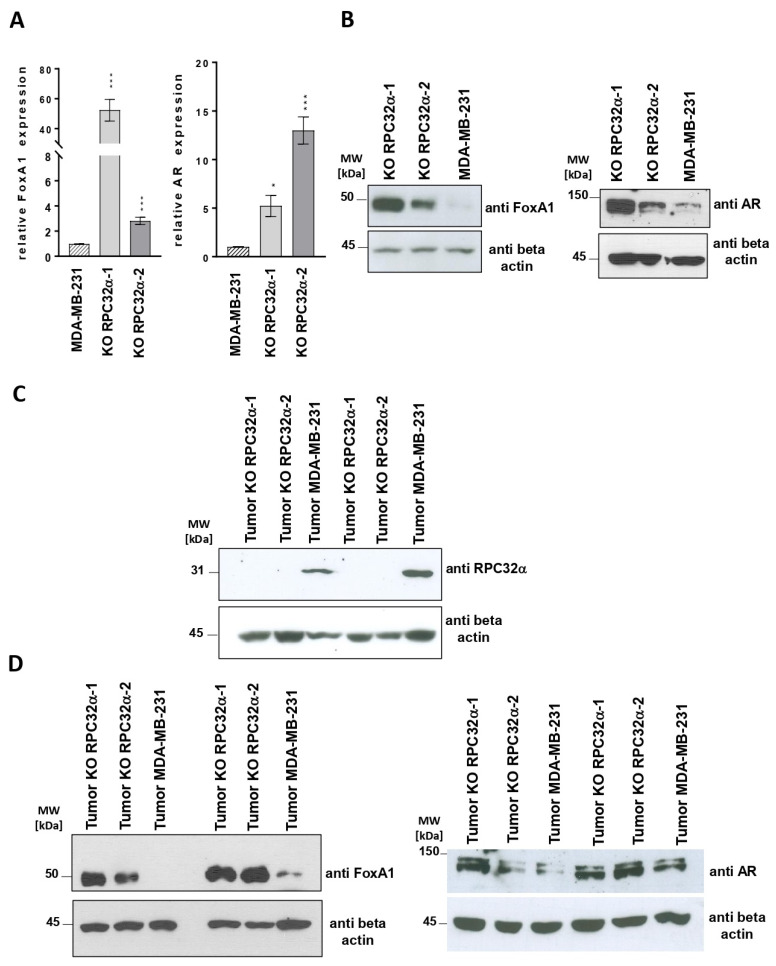
POLR3G/RPC32α KO induces FOXA1 and androgen receptor (AR) expression. (**A**) RT-qPCR analysis of FOXA1 and AR gene expression in MDA-MB-231 and two POLR3G KO (3GKO) cell lines derived therefrom. Mean relative expression of at least three independent experiments ± SEM is shown (* *p* < 0.05, *** *p* < 0.001). The cell lines that were employed are appropriately indicated below the graphs. (**B**) Western blot analyses with anti-FoxA1, -AR and -beta-actin antibodies. Cell lines from which nuclear extracts were derived are appropriately indicated above the ECL images. The proteins that were detected by Western blot are indicated to the right. (**C**) Analyses of RPC32α and beta-actin expression levels by Western blot in resected tumor samples formed by the cell lines that are appropriately indicated above the ECL images. The proteins that were detected by Western blot are indicated to the right. (**D**) Western blot analysis of FoxA1 and AR expression levels in resected tumors formed by the cell lines that are appropriately indicated above the ECL images. The proteins that were detected by Western blot are indicated to the right of each panel. The uncropped Western Blot images of Figure 8B,C can be found in Appendix A, those of Figure 8D in Appendix A.

## Data Availability

The data presented in this study is available within the article or Appendix A.

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
