# Peer review of "The POLR3G Subunit of Human RNA Polymerase III Regulates Tumorigenesis and Metastasis in Triple-Negative Breast Cancer"

_cancers, 2022, doi:10.3390/cancers14235732_

Round 1
Reviewer 1 Report
In this study, the authors investigated the distinct role of POLR3G/RPC32a in triple negative breast cancer (TNBC). They showed POLR3G/RPC32a was clinically significantly overexpressed in TNBC but, neither RNA polymerase iii subunits nor general transcriptional factors were changed. Genomic knockout of POLR3G did not affect primary cell growth however, it severely impaired tumor formation and dissemination both in vitro and in vivo. In conclusion, the authors demonstrated that POLR3G promotes tumorigenesis by regulating genes required for TNBC subtypes such as FOXA1 and androgen receptor (AR). And, they suggest POLR3G as a promising therapeutic target.
Given that RNA polymerase iii is the fundamental enzyme to transcribe short non coding RNAs, it is an interesting study to see whether the enzyme has a potential to modulate tumorigenesis. The authors have clearly proved the importance of POLR3G in TNBC in a variety of ways including clinical data, cellular level and animal models. But, the mechanism by which POLR3G regulates the functions remains elusive. In this regard, it seems necessary to show advisable evidence to support the research.
Questions and comments
Although the authors clearly showed the relevance between highly expressed POLR3G and TNBC, how POLR3G promotes the tumorigenesis remains unknown. In POLR3G KO, there were no changes in RNA polymerase iii-dependently expressed RNAs due to POLR3GL (in Fig. 4C). Considering that RNA polymerase iii is responsible for transcribing short non coding RNAs, it would be better to explain how POLR3G activity modulated the expression of gene sets that are positively or negatively correlated.
POLR3G KO strongly reduced anchorage-independent growth and invasive ability. But, there is no explanation as to why the results were obtained. Comparing global gene expression changes between KO cell line and control cell line could provide meaningful data.
FOXA1 and AR are known as key molecules in luminal breast cancer. I was wondering whether highly expressed FOXA1/AR in POLR3G KO alleviated TNBC like luminal cancer. Please present the experimental evidence whether elevated FOXA1/AR affects the effects or not. If FOXA1/AR mitigates oncogenic characteristics of TNBC, knockdown of FOXA1 or AR in POLR3G KO retrieves oncogenic features of TNBC.
Please show the rescue effect of POLR3G in KO model.
Is it possible to make luminal breast cancer aggressive by forced overexpression of POLR3G?
Author Response
REVIEWER 1
Open Review
(x) I would not like to sign my review report
( ) I would like to sign my review report
English language and style
( ) Extensive editing of English language and style required
( ) Moderate English changes required
( ) English language and style are fine/minor spell check required
(x) I don't feel qualified to judge about the English language and style
|
Yes |
Can be improved |
Must be improved |
Not applicable |
|
|
Does the introduction provide sufficient background and include all relevant references? |
(x) |
( ) |
( ) |
( ) |
|
Are all the cited references relevant to the research? |
(x) |
( ) |
( ) |
( ) |
|
Is the research design appropriate? |
(x) |
( ) |
( ) |
( ) |
|
Are the methods adequately described? |
(x) |
( ) |
( ) |
( ) |
|
Are the results clearly presented? |
( ) |
(x) |
( ) |
( ) |
|
Are the conclusions supported by the results? |
( ) |
(x) |
( ) |
( ) |
Comments and Suggestions for Authors
In this study, the authors investigated the distinct role of POLR3G/RPC32a in triple negative breast cancer (TNBC). They showed POLR3G/RPC32a was clinically significantly overexpressed in TNBC but, neither RNA polymerase iii subunits nor general transcriptional factors were changed. Genomic knockout of POLR3G did not affect primary cell growth however, it severely impaired tumor formation and dissemination both in vitro and in vivo. In conclusion, the authors demonstrated that POLR3G promotes tumorigenesis by regulating genes required for TNBC subtypes such as FOXA1 and androgen receptor (AR). And, they suggest POLR3G as a promising therapeutic target.
Given that RNA polymerase iii is the fundamental enzyme to transcribe short non coding RNAs, it is an interesting study to see whether the enzyme has a potential to modulate tumorigenesis. The authors have clearly proved the importance of POLR3G in TNBC in a variety of ways including clinical data, cellular level and animal models. But, the mechanism by which POLR3G regulates the functions remains elusive. In this regard, it seems necessary to show advisable evidence to support the research.
Questions and comments
Although the authors clearly showed the relevance between highly expressed POLR3G and TNBC, how POLR3G promotes the tumorigenesis remains unknown. In POLR3G KO, there were no changes in RNA polymerase iii-dependently expressed RNAs due to POLR3GL (in Fig. 4C). Considering that RNA polymerase iii is responsible for transcribing short non coding RNAs, it would be better to explain how POLR3G activity modulated the expression of gene sets that are positively or negatively correlated.
- We thank this reviewer for raising this point. We agree that it is very important. However, it will take months or even years to complete. Therefore, we propose that this topic be the subject of future research and not contribute to a revised version of this manuscript.
POLR3G KO strongly reduced anchorage-independent growth and invasive ability. But, there is no explanation as to why the results were obtained. Comparing global gene expression changes between KO cell line and control cell line could provide meaningful data.
- We also agree with the reviewer here. These changes in gene expression caused by the POLR3G KO will without any doubt contribute to a better understanding of how this Pol III subunit causes the observed phenotype. However, we must also note for this point that its realization and analysis will take months or years and will therefore be better suited for future studies rather than contribute to this manuscript.
FOXA1 and AR are known as key molecules in luminal breast cancer. I was wondering whether highly expressed FOXA1/AR in POLR3G KO alleviated TNBC like luminal cancer. Please present the experimental evidence whether elevated FOXA1/AR affects the effects or not. If FOXA1/AR mitigates oncogenic characteristics of TNBC, knockdown of FOXA1 or AR in POLR3G KO retrieves oncogenic features of TNBC.
Please show the rescue effect of POLR3G in KO model.
Is it possible to make luminal breast cancer aggressive by forced overexpression of POLR3G?
- The reviewer made excellent suggestions for the completion of this research project. We fully agree that all of the points, including the last three, would add weight to the manuscript. However, we must point out that these last three experiments would also take months to perform the cloning, cell line establishment and analysis. Therefore, we suggest that these points are more suitable for a future project than for a revision of the manuscript submitted here.

Reviewer 2 Report
In this manuscript Lautre et al, demonstrated specific over expression of POLR3G in triple negative breast cancer. Authors also showed POLR3G knockout in triple negative breast cancer cells reduced metastatic potential and impaired tumor growth. overall the manuscript is intriguing with compelling data and will be interesting to the readers. however, there are some concerns about data representation should be addressed to consider this for publication.
1. Figure 1, color code should be given for the heatmap, which is missing.
2. Figure 3A, is POLR3GL significantly lower in TNBC compared to the normal/other cancer cells?. Quantitation should be done for RPC32 blot. since the difference is not clear from the blot provided. also, RPC7L protein levels should be checked to confirm the lower expression level in TNBC.
4.. Does RPC32 KO alter glycolytic shift in cancer cells?
5. Figure 5, can authors check regulation of some cancer markers either by qPCR or western blot?
6. Figure 6C, better representative image is needed for RPC32 KO panel and also quantitation of IHC staining is required.
Author Response
REVIEWER 2
Open Review
( ) I would not like to sign my review report
(x) I would like to sign my review report
English language and style
( ) Extensive editing of English language and style required
( ) Moderate English changes required
(x) English language and style are fine/minor spell check required
( ) I don't feel qualified to judge about the English language and style
|
Yes |
Can be improved |
Must be improved |
Not applicable |
|
|
Does the introduction provide sufficient background and include all relevant references? |
(x) |
( ) |
( ) |
( ) |
|
Are all the cited references relevant to the research? |
(x) |
( ) |
( ) |
( ) |
|
Is the research design appropriate? |
(x) |
( ) |
( ) |
( ) |
|
Are the methods adequately described? |
(x) |
( ) |
( ) |
( ) |
|
Are the results clearly presented? |
(x) |
( ) |
( ) |
( ) |
|
Are the conclusions supported by the results? |
(x) |
( ) |
( ) |
( ) |
Comments and Suggestions for Authors
In this manuscript Lautre et al, demonstrated specific over expression of POLR3G in triple negative breast cancer. Authors also showed POLR3G knockout in triple negative breast cancer cells reduced metastatic potential and impaired tumor growth. overall the manuscript is intriguing with compelling data and will be interesting to the readers. however, there are some concerns about data representation should be addressed to consider this for publication.
- Figure 1, color code should be given for the heatmap, which is missing.
- An explanation was added to the Figure legend.
- Figure 3A, is POLR3GL significantly lower in TNBC compared to the normal/other cancer cells?. Quantitation should be done for RPC32 blot. since the difference is not clear from the blot provided. also, RPC7L protein levels should be checked to confirm the lower expression level in TNBC.
- We thank the reviewer for this suggestion. We added an anti-RPC32β western blot to Figure 3B.
We propose not to add a quantification of protein levels since western blots are semi-quantitative and all numbers would be at best approximate.
4.. Does RPC32 KO alter glycolytic shift in cancer cells?
- We don’t have data on this. Since this is not a central point of the manuscript, we propose not to include this kind of data.
- Figure 5, can authors check regulation of some cancer markers either by qPCR or western blot?
- We made some preliminary RT-qPCR analyses of EMT genes and did not see any changes. Therefore, we did for the moment not dig deeper into this. We propose not to include these incomplete data into the manuscript. We did not analyze classical breast cancer markers such as TP53, RB1, PTEN and their analysis would be interesting, but would take several months since we would have to determine mRNA and protein levels and analyze their activities. Since we are asked to respond within seven days to the review, we are not able to do these analyses.
- Figure 6C, better representative image is needed for RPC32 KO panel and also quantitation of IHC staining is required.
These data were obtained by platform of experimental pathology/Institut Bergonié Bordeaux (as cited in Materials and methods) who selected the best images and who cannot quantify the IHC staining posteriori.
